# Towards a Sustainable Reproduction Management of Dairy Sheep: Glycerol-Based Formulations as Alternative to eCG in Milked Ewes Mated at the End of Anoestrus Period

**DOI:** 10.3390/ani11040922

**Published:** 2021-03-24

**Authors:** Francesca D. Sotgiu, Cristian Porcu, Valeria Pasciu, Maria Dattena, Marilia Gallus, Giuseppe Argiolas, Fiammetta Berlinguer, Giovanni Molle

**Affiliations:** 1Department of Veterinary Medicine, University of Sassari, 07100 Sassari, Italy; fran.sotgiu@gmail.com (F.D.S.); cporcu@uniss.it (C.P.); vpasciu@uniss.it (V.P.); 2AGRIS Sardegna, Loc. Bonassai, 07100 Sassari, Italy; mdattena@agrisricerca.it (M.D.); mgallus@agrisricerca.it (M.G.); gmolle@agrisricerca.it (G.M.); 3Sementusa^®^, via Torino 13/b, Senorbì, 09040 Cagliari, Italy; kyccco@gmail.com

**Keywords:** dairy ewes insulin, glycaemia, triglycerides, milk, NEFA, urea, pregnancy, ovulation, oestrus, synchronization

## Abstract

**Simple Summary:**

Reproductive management of sheep for autumnal lambing often require induction and synchronization of oestrus and ovulation, either for natural mating or artificial insemination, by the use of pharmacological treatments. Such treatments are mostly based on the administration of progesterone followed by a single intramuscular dose of equine chorionic gonadotrophin (eCG) at progesterone withdrawal. However, repeated eCG treatments in consecutive mating seasons can result in the outbreak of resistance with a rise of anti-eCG antibodies. Furthermore, the future use and availability of eCG appears to be strongly challenged by the highly active animal-rights movement because the hormone is obtained from pregnant mares. The present study demonstrated that the administration of glycerol-based formulations to milked ewes is a valid alternative to eCG treatment in reproductive management protocols based on the induction of ovulation with progesterone-releasing devices at the end of anoestrus period. The glucogenic treatment administration to late lactation dairy ewes at the end of the anoestrus period improved their metabolism without harming animal production or animal welfare, thus promoting a sustainable reproductive management of dairy sheep.

**Abstract:**

This study investigated whether the administration of equine chorionic gonadotrophin (eCG) in a protocol to induce and synchronize ovulations before mating could be replaced by the administration of glycerol-based formulations in milked ewes at the end of their seasonal anoestrus. Forty-eight late-lactation dairy ewes of the Sarda breed were synchronized using sponges impregnated with progestogen and then joined with fertile rams (day (D) 0, ram introduction). From D−4 to D−1, the ewes received by gavage either 100 mL of a glucogenic mixture (70% glycerol, 20% propylene glycol and 10% water; GLU group; *n* = 24) or 100 mL of water (GON group; *n* = 24) twice daily. Moreover, on the day of sponge withdrawal (D−1), GON ewes received 200 IU of eCG. There were no differences in reproductive performances between groups. GLU ewes showed higher glycemia (*p* < 0.001), insulinemia (*p* < 0.05), plasma glycerol (*p* < 0.001), triglycerides (*p* < 0.001) and lower cholesterol (*p* < 0.001), non-esterified fatty acids (NEFA; *p* < 0.05) and urea (*p* < 0.001). Plasma osmolality was higher in GLU but only 4 h after dosing (*p* < 0.001). Milk yield and milk composition were not affected by the treatments with exception of milk glycerol (*p* < 0.001) and milk urea (*p* < 0.001), which were higher and lower in GLU than GON ewes, respectively. In conclusion, the administration of the glucogenic mixture to late lactation dairy ewes at the end of anoestrus period resulted in reproductive responses as good as the ones obtained by the eCG treatment, suggesting that the objective of a sustainable reproductive management of dairy sheep can be successfully pursued.

## 1. Introduction

The reproductive management of sheep for out-of-season lambing may require induction and synchronization of oestrus and ovulation, either for natural mating or artificial insemination by means of pharmacological treatments, which are based mostly on the administration of progesterone or its analogues for mimicking the activity of the corpus luteum. Usually, a single intramuscular dose of equine chorionic gonadotrophin (eCG) is injected at progesterone withdrawal to induce oestrus and ovulation, to reduce variability in the onset of oestrus among treated animals and to increase ovulation as a consequence of a greater follicular development [1]. However, repeated eCG treatments in consecutive mating seasons can result in the outbreak of resistance with a rise of anti-eCG antibodies, lower responsiveness to the drug and hence lower reproductive performance [2]. Hence, there is a need for alternative synchronization protocols without eCG [1,3].

We have recently reported that the administration of glucogenic glycerol-based formulation to Sarda ewes was able to elicit an ovulatory response which did not differ from that of eCG-treated ewes dosed with water [4]. However, the study was conducted during the natural breeding season using a PGF-based synchronization protocol without the administration of progesterone. The short-term use of glucogenic glycerol-based formulations in the ewe’s diet led to a positive metabolic chain of events, starting with a rise in glucose plasma levels [3] which, besides acting as metabolic fuel, also play a role as signalling molecule to stimulate folliculogenesis [1] and determine oocyte quality [5,6]. Previous studies also demonstrated that a single dose of glycerol given at the moment of progesterone-releasing device removal can increase ovulation rates [7,8]. The rise of glucose plasma levels leads in turn to increased circulating concentrations of insulin [3,9,10] and insulin-like growth factor (IGF-1) [10], which often act as signalling molecules between metabolism to fertility [11] by interacting with the reproductive axis at the level of the central nervous system as well as in the follicle. The treatment also lowers the plasma levels of non-esterified fatty acids (NEFA) [3,12,13] and urea [3]. High levels of NEFA and urea have been associated with lower fertility [14,15,16,17] due a change of their concentration in follicular fluid [18,19,20,21] or modification of the uterine environment [15,17]. Thus, the administration of glucogenic glycerol-based formulations at the end of luteal phase prior to ram introduction could stimulate follicular development, leading to ovulation and pregnancy rates comparable to those obtained by a hormonal treatment.

However, a previous study showed that the short-term administration of a high dose of glucogenic mixture (23% on dry matter intake basis) to milked ewes resulted in a temporary reduction of milk yield and milk lactose content and an increase of milk protein and casein. Therefore, considering both these changes in milk yield and composition and the recently reported potential side effects of glycerol on red blood cell indices and plasma osmolality [4], the dose given should be carefully set. In a previous study we reported that medium doses of glucogenic mixtures (≈12% on dry matter intake basis) should be preferred for flushing dairy ewes, as they proved to be effective at metabolic level without causing alterations in RBC indices or possibly in their functionality [4].

The present study arose from this finding. Keeping in mind the above premises, it was aimed at (1) investigating whether the administration of eCG in a standard synchronization protocol could be replaced by the administration of glycerol-based formulations in milked ewes at the end of their seasonal anoestrus; (2) monitoring the changes in lipid profile, liver and kidney function consequent to glycerol administration; (3) assessing whether the dose set in a previous study; and [4] preserving milk yield and composition. Moreover, given the high diffusibility of glycerol across plasma membranes, the concentrations of milk glycerol were also determined.

## 2. Materials and Methods

### 2.1. Animals and Treatments

The experiment was run during May–June 2019 at Bonassai research station of Agris Sardegna (40° N, 8° E, 32 m above sea level) at the end of the anoestrus season described for Sarda breed at this latitude. Forty-eight adult (mean ± s.d., 4.5 ± 1.4 years) and multiparous lactating Sarda dairy ewes were used. They were selected from the experimental flock after two consecutive ovarian ultrasound scans performed 8 days apart confirmed the absence of corpora lutea suggesting that ewes were in their seasonal anoestrus period. On the day of the second ultrasound examination, the ewes were divided into two experimental groups homogeneous for body weight (BW), body condition score (BCS), days in milk (DIM) and milk yield (MY; g/day): glucogenic treated group (GLU, *n* = 24; BW 44.7 ± 3.3 Kg; BCS 2.53 ± 0.08, DIM 177 ± 6.1; MY 806 ± 240 g); gonadotrophin treated group (GON, *n* = 24; BW 44.7 ± 4.1 Kg, BCS 2.53 ± 0.08; DIM 176.5 ± 5.6; MY 826 ± 222 g). Each group was further divided in two subgroups, which were used as replicates.

The experimental protocol is depicted in Figure 1, where day 0 (D0) is the day of ram introduction. In brief, oestrus synchronization was induced in all the animals with the insertion (D−13) of one intravaginal progestagen-impregnated sponge (45 mg fluorogestone acetate, FGA, Chronogest; Intervet International, Boxmeer, the Netherlands) which remained in situ for 11 d. On the day of sponge withdrawal (D−2), GON ewes received an intramuscular injection of 200 IU of eCG (Folligon, MSD Animal Health Srl, Segrate, Italy). From D0 to D2, all ewes were joined with fertile rams (*n* = 10) fitted with crayon markers.

From D−4 to D−1, the GLU group received 100 mL of a glucogenic mixture orally twice daily at 0800 h and at 1900 h. The dose of the glucogenic formulation was chosen on the basis of the results of a previous study that compared 8 different doses of glucogenic formulation [4].

The glucogenic formulation administered contained 70% glycerol, 20% propylene glycol and 10% water. Glycerol and propylene glycol had a purity grade of 99.5–100% and complied with EU Reg. 231/2012 for food additives (E422 and E1520 for glycerol and propylene glycol, respectively; Farmalabor srl, Assago, Milano, Italy). The amount of energy supplied was 0.6 NE_L_ Mcal/d, corresponding to ca. 12% of the expected intake of offered diet on DM basis. Gonadotrophin treated ewes received 100 mL of water twice daily simultaneously to treatment administration. Both the glucogenic formulation and the water were administered using a drench gun.

From D−13 to D+2 (i.e., throughout the treatment periods), the ewes of both groups were kept indoors in separate pens and were machine-milked twice daily at 0700 h and 1500 h.

Indoor daily feeding for both groups consisted of 600 g/head of a commercial pelleted feed divided into two equal meals at milkings fed individually in the milking parlour: 1000 g/head of dehydrated lucerne hay and 1000 g/head of a ryegrass hay fed in different troughs in two meals after each daily milking. On the blood sampling day, the morning meal based on the pelleted concentrate was administered immediately after the first blood sampling. Water and mineral-vitamin blocks were available ad libitum.

### 2.2. Feedstuff Composition

Samples of the hays and the concentrate were collected every week and bulked to form two samples per feed for subsequent determinations. All these samples were oven-dried at 65 °C and subsequently ground to pass a 1 mm screen to determine the content of dry matter (DM, oven drying at 100 °C overnight), ash (ID#942.05), ether extract (EE, ID# 920.39) and CP (N × 6.25, [ID# 988.05]) according to AOAC [22]. NDF, acid detergent fiber on an ash-free basis (ADF) and acid detergent lignin (ADL) were also determined [23]. Finally, the in vitro dry matter digestibility (IVDMD) of all feed was measured by the pepsin–cellulase method [24] whereas the starch content of the concentrate was assessed by polarimetry. Net Energy (NE_L_, Mcal/kg DM) and metabolic protein content of feedstuffs and diet were calculated using the equations published by Cannas et al. [25]. Diet formulation was done using Small Ruminant Nutrition System package. Feedstuff chemical composition is shown in Table 1. Estimated average NE_L_ were 1.60 Mcal/kg DM, (concentrate), 1.00 Mcal/kg DM (dehydrated lucerne), 0.80 Mcal/kg DM (grass hay), and 2.70 Mcal/kg DM (glucogenic mixture).

### 2.3. Blood Samplings

Before starting the glucogenic treatment (D−5 of the experimental period), blood samples were collected at fasting (0800 h). In addition, on the third day of glucogenic treatment administration (D−2 of the experimental period), two blood samples were collected, one at fasting immediately before the morning administration of the glucogenic formulation at (0800 h) and one four hours later (1200 h). Finally, three days after the end of the glucogenic treatment a final blood sample was collected at fasting (D2; 0800 h).

At each sampling, from each ewe, two blood samples were collected: one using 2-mL vacuum collection tubes with glycolytic inhibitor (5.0 mg sodium fluoride, 4.0 mg potassium oxalate—Vacutainer Systems Europe; Becton Dickinson, Le Pont-de-Claix, France) for glucose assay; one using 2.0 mL vacuum collection whole blood tube with spray-coated K_2_EDTA (Vacutainer Systems Europe; Becton Dickinson, Le Pont-de-Claix, France) for metabolites and insulin quantification. Immediately after recovery, samples were cooled to 4 °C. Blood samples were centrifuged at 1500× *g* for 15 min at 4 °C degrees. Individual plasma was removed and stored in vials at −20 °C until assayed.

### 2.4. Plasma Osmolality Determination

Plasma osmolality (Osm/kg) was measured using a freezing point osmometer (Osmomat 030, Gonotec, Berlin, Germany).

### 2.5. Metabolites

Plasma samples were measured in duplicate. Glycerol concentration was measured in a single assay by colorimetric method using a commercial free glycerol assay kit (Cell Biolabs, Inc., San Diego, CA, USA), with glycerol standards in the concentration range of 0–400 μM. The kit measures free, endogenous glycerol by a coupled enzymatic reaction system. The glycerol was phosphorylated and oxidized to produce hydrogen peroxide, which reacted with the kit’s colorimetric probe (absorbance maxima of 570 nm). The analytical detection limit was 5 μM.

Glucose, NEFA, Urea, total cholesterol and triglycerides, creatinine, total protein, albumin, alanine aminotransferase (ALT) and aspartate aminotransferase (AST), were measured using commercial kits (Real Time Diagnostic Systems kits) and BS-200 Mindray (Adaltis, Milan, Italy) clinical chemistry analyzer. We used Serum I Normal (Wako) and Serum II Abnormal (Wako) as a multicontrol for each measured parameter. Glucose concentrations were determined in a single assay by the liquid enzymatic colorimetric method (GOD-POD) with a glucose standard of 100 mg/dL for calibration. Intra-assay CV value was 1.1%. NEFA, Urea, total cholesterol and triglyceride concentrations were measured in multiple assays by the enzymatic endpoint method with different standards for calibration: 1 mmol/L, 50 mg/dL, 200 mg/dL and 2.28 mol/L for NEFA, urea, total cholesterol and triglycerides respectively. The intra-assay and interassay CV values were 1.07 and 0.98% (for NEFA), 1.7 and 1.6% (for urea), 0.95 and 1.24% (for total cholesterol) and 0.99 and 1.05% (for triglycerides) respectively. Creatinine was measured in an assay by the increasing kinetics method with a standard of 2 mg/dL for calibration; its intra-assay and interassay CV values were 1.07 and 2.55%, respectively. Total protein and albumin concentrations were measured in multiple assays by spectrophotometric endpoint methods, with the following standards for calibration: 6 g/dL and 46.2 g/L of bovine albumin for total protein and albumin assays respectively. Their intra-assay and interassay CV values were 1.94 and 1.35% for total protein and 1.36 and 1.52 for albumin, respectively. ALT and AST concentrations were measured in multiple assays by kinetics UV method using an ALT standard solution of 97.65 U/L and AST standard of 102 U/L for calibration. Their intra-assay and interassay CV values were 2.07 and 2.24% (for ALT) and 2.42 and 2.29% (for AST) respectively.

### 2.6. Insulin

ELISA assays were performed using the Personal Lab Adaltis (Adaltissrl, Rome, Italy), a tool that performs automated ELISA protocols. Insulin concentration was measured in duplicate using a commercial ovine insulin ELISA Kit (Mercodia developing diagnostics, Marburg, Germany) which is a solid-phase ELISA based on the direct sandwich technique. The kit is calibrated against an in-house reference preparation of ovine insulin, and it has been previously used for insulin determination in ovine plasma [3,4]. The mean ovine insulin concentrations of the six reference solutions were 0, 0.05, 0.15, 0.5, 1.5, and 3 mg/L. The recovery on addition was 94%–114% (mean 103%). The analytical sensitivity was 0.025 mg/L and the intra-assay and interassay CV values were <7%.

### 2.7. Intake, Body Weight, Body Condition Score, Milk Yield and Milk Composition

Feed intake at group level was measured by weighing the feed offered and the corresponding orts after each meal for the concentrate and after 24 h for the hays. Body weight was measured before the morning meal using an electronic scale before (D−13) and after the glucogenic treatment (D2). On the same days, body condition score ranging from 1 (extremely thin) to 5 (obese) was estimated by two trained evaluators with an approximation of 0.25 BCS units [26]. Their scores were averaged prior to data analysis.

Milk yield was measured by weighing the production of each ewe at two consecutive milkings at 0700 h and at 1500 h on two occasions before (D−10 evening and D−9 morning) and during (D−3 evening and D−2 morning) glucogenic treatment. Milk samples were collected and milk composition assayed on composite samples for fat, protein, casein and lactose using the Fourier-transformed infrared method (Milkoscan FT+, Foss Electric, Hillerød, Denmark) and milk urea concentration using an enzymatic colorimetric assay (Chem Spec 150; Bentley Instruments Inc., Chaska, MN, USA). The glycerol concentration in the milk was measured as above described for blood.

### 2.8. Ultrasound Scanning

Ovarian ultrasound scanning was performed in the pre-experimental period and on D10 of the experimental period to determine ovulation rates by counting the corpora lutea present in each ovary. In both cases, the ovaries were examined by transrectal ultrasonography using a real-time, B-mode scanner (Aloka SSD 500; Aloka Co. Ltd., Tokyo, Japan) fitted to a 7.5 MHz linear-array probe (82 mm prostate transducer UST-660-7.5, Aloka Co.). In each observation, each ovary was scanned several times from different angles to determine the presence and the number of all corpora lutea. Pregnancy diagnosis was performed on D40 using transrectal ultrasonography (Aloka SSD 500, fitted to 82 mm prostate transducer UST-660-7.5, Aloka Co.). Pregnant sheep displayed enlargement of uterine horns and an embryo heartbeat was evident.

### 2.9. Statistical Analyses

Differences between groups in body weight and BCS and their changes during the glucogenic-treatment period were analysed by a mono-factorial general linear model (GLM).

Mean plasma circulating concentration of metabolites and osmolality on day −2 (during treatment period), milk glycerol, milk yield and composition were analysed by GLM with time, treatment groups and their first-order interactions as fixed effects using Minitab 17 Statistical Software (2010, Minitab, Inc., State College, PA, USA). As post-hoc test, Tukey’s test was used to highlight differences within and between groups.

A chi-square analysis was used to highlight the differences between groups for ovulation and pregnancy rates.

Results are expressed as mean values (mean ± SE) and the differences were considered to be statistically significant at *p* < 0.05.

## 3. Results

At the end of the treatment period, no differences were observed in body condition between the experimental groups (Appendix A). During the treatment period, the concentrate and the lucerne dehydrated hay were almost completely consumed in both groups (525 and 523 g DM/head, for the concentrate and 797, 795 g DM/head for the lucerne, in GON and GLU groups, respectively), with no evidence of abrupt changes during treatment period. In contrast, the intake of ryegrass hay (537 and 499 g DM/head) tended to decrease more in GLU than GON group in the 4 days of glucogenic administration as compared with the previous 4 days (−9% vs. −22%, in GON and GLU, respectively, one-tail student t test for the difference between GON and GLU after the beginning of the treatment, *r* = 8, T = 2.12, *p* = 0.07).

Regarding the reproductive performances, no significant differences in ovulation and pregnancy rates were found between glycerol- and gonadotrophin-treated ewes (Table 2).

The metabolic status of the GLU group was modified compared to the GON group, showing a more favourable energy balance. As expected, the glucogenic administration led to a rise in glycerol plasma levels on GLU group (*p* < 0.001; Table 3), and consequently of plasma osmolality (*p* < 0.01; Table 3) as shown in a previous study [4]. On the third day of glycerol-based formulation administration (D−2), GLU group showed higher glycemia 4 h after the morning treatment (*p* < 0.001; Table 3), and insulinemia at fasting and after treatment administration (*p* < 0.05; Table 3) compared to the GON group. Moreover, on the same day, these effects reflected on lower plasma levels of NEFA at fasting (*p* < 0.05; Table 3) and lower urea mean circulating concentrations (*p* < 0.001; Table 3). This metabolic shift included a significant rise in the circulating concentration of triglycerides (Table 3) in the GLU group, whose values were 15-fold higher than those found in gonadotrophin treated ewes (*p* < 0.001).

Milk yield was not affected by the nutritional treatment (Table 4), while milk composition was modified in the urea concentration, which reflected the changes observed in the plasma (Table 4). It is noteworthy that glycerol proved to reach the mammary gland, as demonstrated by the significant higher glycerol milk concentration found in the GLU group compared to the GON one (6.480 ± 1.050 and 0.497 ± 0.193 mg/mL, in GON and GLU groups, respectively; *p* < 0.001).

## 4. Discussion

Research efforts are focusing on finding alternative treatments to eCG administration in protocols to induce oestrus and ovulation in small ruminants [4,27,28] considering the concerns linked to its production for brood mare health and welfare [29]. The reduction in the use of synthetic hormones in livestock farming would also meet consumer demand for “natural, green, and clean” methods [30].

The present study demonstrated that ovulation and pregnancy rates did not differ in glucogenic- and eCG-treated ewes. Thus, the administration of a glycerol-based glucogenic formulation can replace hormonal stimulation with eCG in a standard protocol with progesterone-releasing devices for the induction of oestrus and ovulation in milked Sarda ewes at the end of their seasonal anoestrus. This finding confirms previous findings in dry ewes subjected to a prostaglandin-based synchronization period during the breeding season [4]. However, pregnancy rates were below expectations considering that rates as high as 70% are usually achieved [31]. A slightly better pregnancy rate (0.60) was achieved in a previous experiment in which a double dose of glucogenic formulation was administered to milked ewes [3], but in that study the ewes were in a better condition at mating (BCS = 2.7 vs. 2.5). Moreover, in this study as well as in the previous one, the sheep could have suffered from heat stress. Actually, just after mating there were at least two days in June with average THI > 72 (severe heat stress according to, Marai et al. [32]) with more than 8 h/daily above 30 °C. These conditions can favour embryo mortality [32]. Preliminary results of our laboratories suggest that high doses of glucogenic (i.e., the double of the one used in this experiment) can increase eye and rectal temperature by almost 1 C° in ewes experiencing severe heat stress [33]. This aspect warrants further investigation in the light of the increasing risks of dramatic heat waves as a consequence of the climate change.

Glucogenic formulations can boost ovarian activity by creating a suitable systemic and intra-follicular metabolic milieu for the promotion of ovarian function [10]. The dose of the glycerol-based formulation used in the present study had been previously designed [4] to ensure this positive switch in ewe metabolic status without causing any significant alteration in red blood cell (RBC) indices. Higher doses (from 18.2 to 27.4% on DM basis) cause indeed significant changes in RBC indices consequent to glycerol diffusion across the RBC membrane, which may be associated with impaired RBC functionality [34,35]. However, our previous study [4] was carried out in dry ewes during their breeding season. The present study thus confirms that this dose was able to elicit a comparable metabolic status also in milked ewes at the end of their seasonal anoestrus although during lactation, lipolysis and proteolysis are increased to meet the energetic requirements of the mammary gland.

Moreover, the more complete metabolic status assessment performed in the present study revealed that the administration of the glycerol-based formulation, while not significantly affecting liver and kidney functions, led to a significant rise in circulating triglyceride concentrations. Under negative energy balance (NEB), which is frequently found in ruminants in the peripartum period or during lactation [36,37], NEFAs were released from the triglycerides stored in adipose tissue by lipases that split them into NEFA and glycerol following a reduction in blood glycemia. Once in the hepatocytes, NEFAs enter the mitochondria and underwent β-oxidation to produce energy as an alternative energy source to glucose. However, when a positive shift in energy balance occurred and glycemia increased, there was a subsequent rise in insulinemia-stimulated glucose uptake and glycolysis by the hepatocytes. This reaction produced glycerol-3-phosphate, which acted as a substrate for the re-esterification of NEFAs to triglycerides [38]. Thus, the significant rise in circulating triglyceride concentrations, together with the significant decrease in circulating NEFA concentrations, in glycerol-treated ewes confirmed that the administration of the glycerol-based formulation led to a switch from fat oxidation to carbohydrate utilization [39].

The significant rise in circulating triglyceride concentrations consequent to the administration of the glycerol-based formulation may have also been triggered by the increase in the plasma glycerol concentrations. When the circulating glycerol concentration exceeded its baseline level, its inclusion in hepatic gluconeogenesis and triglyceride synthesis increased [40,41]. In the GLU group glycerol metabolism shifted indeed to the liver synthesis of fatty acids and then triglycerides. In humans, treatments with glycerol have been associated with an increase in hypertriglyceridemia [42]. A similar effect was reported in rats [43]. On the other hand, the inclusion of crude glycerin in the diets for growing lambs caused no significant increase in triglyceride plasma concentrations [44,45]. This difference may be related to differences in metabolism between growing and lactating sheep and to the significant decrease in dry matter intake found in lambs following glycerol inclusion in the diet [44,45]. However, in the present study, mean cholesterol concentrations were lower in glucogenic-treated compared to eCG-treated ewes. Similar results (high triglyceride and low cholesterol) were found by Mazur et al. [46] in well-fed compared to underfed pregnant ewes. In sheep, as well as in other ruminants the level of lipoprotein in blood is usually lower than in monogastric mammals, with a lower secretion of VLDL-triglyceride in ruminant liver and hence higher risk of steatosis at the end of pregnancy [46]. However, in the case of this study, the rise was detectable only in the post-treatment blood sampling. Moreover, since the glucogenic treatment was limited to 4 days, this period was probably too short to cause the above-mentioned lipid alterations.

The effect of high triglyceride concentrations on ovarian activity was poorly explored. However, triglycerides did not pass through the follicular membrane [47] and their intracellular concentration was mainly the result of local metabolic processes [19]. Therefore, an acute rise in triglycerides should not affect follicular development.

The lower dose of glucogenic mixture used in the present study caused no significant reduction in milk production and lactose contents, thus backing the opportunity of reducing the dose administered. In contrast, the significant decrease in milk urea in glycerol-treated compared to eCG-treated ewes reflected the positive change in the metabolic status already pointed out. Moreover, high milk urea is a marker of reduced fertility in lactating cows [48] and ewes [49]. The significant higher milk glycerol concentration consequent to the administration of the glycerol-based formulation suggests that glycerol from the bloodstream can pass through the mammary gland. The presence of glycerol in milk could affect some physical and chemical characteristic [50]. Goats orally treated at increasing levels with bi-distilled glycerol [51] showed that glucogenic treatment could affect cheese softness. When the treatment is made on few animals (individual cows in the post-partum period) the quantity of glycerol on bulk milk can be negligible. However, in small ruminants the effect on milk quality should be taken into account when the treatment is administered to the entire flock during the mating season, particularly when higher doses of glycerol are given or the administration lasts for a longer time.

## 5. Conclusions

In conclusion, the present study demonstrated that the administration of glycerol-based formulations to milked ewes can be a valid alternative to eCG treatment in reproductive management protocols based on the induction of ovulation with progesterone-releasing devices at the end of anoestrus period at our latitudes. This finding confirmed previous results in dry ewes submitted to the same treatment during the breeding season, thus suggesting that the objective of meeting consumer demand of natural “green, clean and ethical products” [30] can be successfully pursued. However, given the diffusion of glycerol in the mammary gland, further investigations are needed to study the effects of the treatment on milk composition and the possible influence on cheesemaking. Moreover, more information should be gathered on the effects of glucogenic dosing when mating occurs under extreme heat conditions.

## Figures and Tables

**Figure 1 animals-11-00922-f001:**
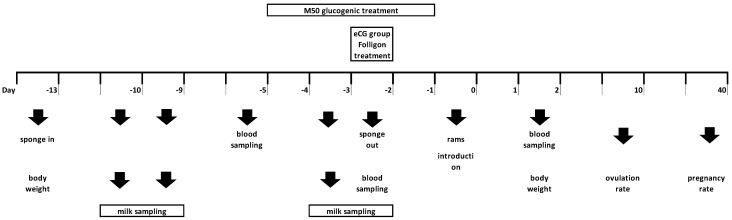
Experimental protocol—Figure shows exact timing at which treatments, samplings and measurements were performed; eCG- equine chorionic gonadotrophin.

**Table 1 animals-11-00922-t001:** Mean and standard deviation of feed components. *n* = 2 samples per feedstuff.

	Concentrate		Grass Hay		Dehydrated Lucerne Hay	
	Mean	s.d.	Mean	s.d.	Mean	s.d.
DM	87.94	0.65	87.54	0.23	89.95	0.49
Ash	8.66	1.16	13.50	7.43	10.28	0.30
EE	3.36	0.12	1.42	0.25	1.85	0.01
CP	15.43	1.29	7.49	1.65	15.04	0.03
NDF	23.68	3.39	61.95	12.26	49.53	0.30
ADF	11.41	1.99	39.81	9.53	36.47	0.22
ADL	2.25	0.41	6.45	0.35	8.11	0.21
Starch	36.65	6.33	nd		nd	
IVDMD	85.10	2.36	45.37	9.84	59.68	1.98

DM: dry matter; EE: ether extract; CP: crude protein; NDF: neutral detergent fiber; ADF: acid detergent fiber on an ash-free basis; ADL: acid detergent lignin; IVDMD: in vitro dry matter digestibility.

**Table 2 animals-11-00922-t002:** Ovulation and pregnancy rates in lactating ewes mated during their seasonal anoestrus period after a synchronization protocol based on progesterone-releasing devices and an oral drench with a glycerol-based formulation (GLU; *n* = 24) or water but receiving an eCG administration (GON; *n* = 24).

Treatment	Ovulation Rate			Pregnancy Rate		
	Mean		SE	Mean		SE
**GLU**	1.52	±	0.15	0.52	±	0.11
**GON**	1.33	±	0.12	0.57	±	0.11

**Table 3 animals-11-00922-t003:** Plasma osmolality and circulating concentrations of metabolites in ewes orally drenched with a glycerol-based formulation (GLU; *n* = 24) or with water but receiving an eCG administration (GON; *n* = 24) before (T0) and 4 h after (T4) the drenching on day 3 of the treatment period (D−2). Lowercase letters indicate significant differences between the two time points within GLU group: osmolality *p* < 0.001; glycerol *p* < 0.001; glucose *p* < 0.001; insulin *p* < 0.05; triglycerides *p* < 0.001. Uppercase letters indicate significant differences between the two time points within the GON group: NEFA *p* < 0.05. Asterisks indicate significant differences between groups at the same time point: osmolality *p* < 0.01; glycerol *p* < 0.001; glucose *p* < 0.001; insulin *p* < 0.05; NEFA *p* < 0.05; urea *p* < 0.001; triglycerides *p* < 0.001; cholesterol *p* < 0.001; AST *p* < 0.05; albumin *p* < 0.001. (GLM—Tukey’s post hoc test).

		Treatment		*p* Values		
		GLU	GON	Groups	Time	Groups*Time
Osmolality (Osm/kg)	D−2 T0 h	0.301 ± 0.002 ^a^	0.307 ± 0.002	0.221	0	0.001
	D−2 T4 h	0.324 ± 0.003 ^b^*	0.312 ± 0.002 *			
Glycerol (mg/mL)	D−2 T0 h	0.578 ± 0.009 ^a^	0.511 ± 0.018	0	0	0
	D−2 T4 h	23.520 ± 5.120 ^b^*	0.573 ± 0.009 *			
Glucose (mg/dL)	D−2 T0 h	50.504 ± 0.692 ^a^	50.550 ± 1.050	0	0	0
	D−2 T4 h	84.810 ± 4.170 ^b^*	51.900 ± 1.370 *			
Insulin (µg/L)	D−2 T0 h	0.529 ± 0.126 ^a^*	0.128 ± 0.023 *	0.048	0.004	0
	D−2 T4 h	1.001 ± 0.121 ^b^*	0.219 ± 0.021 *			
NEFA (mmol/L)	D−2 T0 h	0.112 ± 0.005 *	0.147 ± 0.009 ^A^*	0.03	0.003	0.014
	D−2 T4 h	0.106 ± 0.007	0.109 ± 0.008 ^B^			
UREA (mg/dL)	D−2 T0 h	41.650 ± 2.010	52.470 ± 2.440	0.319	0.032	0
	D−2 T4 h	44.400 ± 2.300	59.880 ± 2.540			
Triglycerides (mg/dL)	D−2 T0 h	8.149 ± 0.379 ^a^	16.323 ± 0.831	0	0	0
	D−2 T4 h	234.500 ± 37.300 ^b^*	15.103 ± 0.920 *			
Cholesterol (mg/dL)	D−2 T0 h	44.880 ± 1.410	49.060 ± 1.030	0.919	0.022	0.001
	D−2 T4 h	42.180 ± 1.380	46.110 ± 0.947			
ALT (U/L)	D−2 T0 h	25.510 ± 1.200	26.320 ± 0.992	0.85	0.674	0.57
	D−2 T4 h	26.160 ± 1.150	26.566 ± 0.891			
AST (U/L)	D−2 T0 h	95.220 ± 2.460	102.250 ± 2.260	0.515	0.308	0.024
	D−2 T4 h	99.220 ± 2.480	103.140 ± 2.330			
Albumin (g/dL)	D−2 T0 h	3.013 ± 0.031	3.110 ± 0.037	0.6	0.166	0.001
	D−2 T4 h	2.946 ± 0.038	3.080 ± 0.033			
Total protein (g/dL)	D−2 T0 h	7.272 ± 0.093	7.315 ± 0.093	0.856	0.067	0.525
	D−2 T4 h	7.079 ± 0.103	7.157 ± 0.091			
Creatinine (mg/dL)	D−2 T0 h	0.846 ± 0.016	0.855 ± 0.019	0.726	0.209	0.834
	D−2 T4 h	0.831 ± 0.015	0.829 ± 0.016			

NEFA: Non-Esterified Fatty Acids; ALT: alanine transaminase; AST: Aspartate transaminase.

**Table 4 animals-11-00922-t004:** Milk yield and composition in ewes orally drenched with a glycerol-based formulation (GLU; *n* = 24) or with water but receiving an eCG administration (GON; *n* = 24) before and during the treatment period (D−10/D−9; D−3/D−2). Lowercase letters indicate significant differences between the two time points within the GLU group: urea *p* < 0.01. Asterisks indicate significant differences between groups at the same time point: urea *p* < 0.001 (GLM—Tukey’s post hoc test).

		Treatments		*p* Values		
		GLU	GON	Groups	Days	Groups*Days
Milk yield (L)	D−10/D−9	0.753 ± 0.054	0.791 ± 0.033	0.76	0.224	0.603
	D−3/D−2	0.720 ± 0.054	0.710 ± 0.041			
Milk fat (g/100 g milk)	D−10/D−9	7.176 ± 0.198	7.023 ± 0.275	0.668	0.002	0.731
	D−3/D−2	6.485 ± 0.167	6.468 ± 0.118			
Protein (g/100 g milk)	D−10/D−9	5.883 ± 0.139	5.661 ± 0.210	0.2	0.987	0.895
	D−3/D−2	5.865 ± 0.160	5.684 ± 0.089			
Lactose (g/100 g milk)	D−10/D−9	4.210 ± 0.102	4.257 ± 0.144	0.494	0.127	0.769
	D−3/D−2	3.988 ± 0.123	4.106 ± 0.111			
Somatic cells (log 10 SCC/mL)	D−10/D−9	2.582 ± 0.104	2.543 ± 0.087	0.31	0.414	0.539
	D−3/D−2	2.722 ± 0.106	2.563 ± 0.091			
Casein (g/100 g milk)	D−10/D−9	4.432 ± 0.121	4.308 ± 0.165	0.327	0.806	0.983
	D−3/D−2	4.403 ± 0.135	4.273 ± 0.082			
Urea (mg/dL)	D−10/D−9	44.300 ± 1.670 ^a^	41.310 ± 2.020	0.085	0.002	0.001
	D−3/D−2	33.000 ± 1.440 ^b^*	41.810 ± 1.490 *			

## Data Availability

The data presented in this study are available on request from the corresponding author. The data are not publicly available due to a temporary lack of a public accessible repository.

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
