# Peer review of "Towards a Sustainable Reproduction Management of Dairy Sheep: Glycerol-Based Formulations as Alternative to eCG in Milked Ewes Mated at the End of Anoestrus Period"

_animals, 2021, doi:10.3390/ani11040922_

Round 1
Reviewer 1 Report
The authors addressed all suggestions recommended by this reviewer. In our opinion, this version is suitable for publication.
Reviewer 2 Report
The authors have made a great effort and have revised clearly the manuscript and have made all the changes suggested by this reviewer and have incorporated all the points and all the edits indicated. Now, they have really produced an excellent manuscript.
This is a fantastic manuscript and the authors should be commended for produced such high-class studies - excellent!
The manuscript can now be accepted as it is.
This manuscript is a resubmission of an earlier submission. The following is a list of the peer review reports and author responses from that submission.
Round 1
Reviewer 1 Report
General
The authors try to explain facts well-known in everybody working with small ruminants, facts known even to elder Italian shepherds, therefore, regarding novelty there is limited scope for this manuscript. However, the authors have used modern techniques, which have added value to the study, rendering the work more up to date.
Before potential acceptance, there is a need for shortening and significant corrections in the manuscript, to make it suitable for publication.
Introduction
The section is particularly long and should be reduced. The first three paragraphs present well-known facts and should be deleted.
Objectives of the work not described.
Materials and methods
The experimental design needs to be described in brief in a table.
As much as possible of the samplings and examinations made, should be included in that table, by decreasing proportionately the length of the manuscript. This should be included as supplementary material.
In general, please transfer as supplementary material as much as possible from M&M.
Results
Figures 2 and 4 to be transferred to supplementary material. Moreover, relevant tables should also be included. Figures only show trends, not real results.
Figures of the ultrasound of the ovaries taken at 10 days post-conception MUST BE INCLUDED by any means. Lack of inclusion will be interpreted that the work was not performed.
Discussion
The discussion is long and verbose. Whilst the concerns of the consumers have been voiced indeed, the issue has been overplayed and scientists should not be lured by voices of activists. I suggest to downplay the discussion, staying only in the scientific facts, without much political noise. Further, the authors should write that there is anecdotal evidence about the effects of glycogenic factors throughout Italy and this work tries to explain the known facts and to establish a scientific basis behind them.
References
Important Italian references in the same narrow field are missing and must be added.
Overall, the manuscript is useful only because it explains, not because it describes for the first time and this must be written clearly in the final section.
Author Response
Dear Reviewer,
please find Author's reply in the attached file.
Best regards,
Fiammetta Berlinguer

Reviewer 2 Report
My advice is to add "of glucogenic mixture" in line 396 after: the lower dose.
Author Response
Dear Reviewer,
we modified the manuscript following your suggestion:
Point 1: My advice is to add "of glucogenic mixture" in line 396 after: the lower dose
Response 1: we modified the sentence as suggested and added “the lower dose of glucogenic mixture used in the present study …” (line 410).
Thank you for improving the quality of our work.
Best regards,
Fiammetta Berlinguer
Reviewer 3 Report
General comments:
This study aimed to evaluate if short-term glycerol dietary supplementation can be an alternative to eCG administration to induce ovulation followed by pregnancy in native (Sarda) ewes at the end of the anestrous season. The manuscript is well written, but several points need to be addressed. The lack of a control group is a major problem to reach the objective of the study. In fact, great part of the manuscript (except the introduction), and mainly the discussion is dedicated to the metabolic finding between both tested groups. Probably, the objectives and title of the manuscript need to be corrected to mitigate this aspect. The eCG group can serve as the control group to study the effect of Glycerol on the studied metabolites.
A second largest limitation is the conditions of the experiment: We don’t know if the ewes (or what percentage) are cycling or in anestrous. The study protocols are based on 11 days of P4 priming plus rams introduction on day 13, determination of the ovulation number and pregnancy (by trans-rectal ultrasonography) on day 23 and 40, respectively. This means that the ewes (or a percentage) could be previously cycling), a natural luteolysis occurred, and the (anestrous) ewes could have been stimulated by the male effect. The authors need to clarify these points.
Specific comments:
L86-87: I suggest changing “… PGF-based synchronization protocol.” to “… PGF-based synchronization protocol without administration of progesterone.”
L87: I suggest changing “The use of glucogenic…” to “The short-term use of glucogenic…”
L118-119: milk yield- g/day
L124: According to Fig. 1, the sponges were removed at D-2.
L125: A breeding score examination is previously performed? How these rams were separately confined from ewes? I.e., can be possible an additional male effect to interact with the output results (ovulation and pregnancy)?
L138-139: “The amount of energy supplied by the glucogenic formulation was….”. can the authors give more details about the calculation of the estimated 13% NEL?
L232: D45 or D40?
L255: The authors mean P=0.07?
L336: In my opinion, the authors cannot use the term “demonstrated” once the study design doesn’t consider a control group (without eCG administration or Glycerol supplementation), but only a comparison between ewes receiving eCG or Glycerol. This study only demonstrates that no differences in ovulation and pregnancy rates between the two treated groups under the conditions of the experiments. It can be suggested that no differences in the studied reproductive output can occur between groups under these conditions. Please see the general comments.
L338-339: I suggest changing “…in milked ewes…” to “…in milked Sarda ewes…”
L340-341: The ewes of the present study were cycling?
L366: Please change the reference [42] regarding ruminants.
L371: The described metabolism is a normal process in ruminants. So, please correct the term “… speculate…”
L372-383: This part should be more carefully described. As the authors know there are important differences between ruminants and monogastric species.
L394-395: Can the effect of Glycerol on ruminal microbiota and the proportion of volatile fatty acids explain partially this fact?
L422-424: I suggest “In conclusion, the present study suggest that the administration of glycerol-based formulations to milked Sarda ewes can be a valid alternative to eCG treatment in reproductive management protocols based on the induction of ovulation with progesterone-releasing devices at the end of anestrous period reared in our latitudes”; and, “Further research is needed to confirm these results in confirmed anestrous and cycling ewes at this time as well considering other indigenous breeds at similar latitude.” or similar.
Author Response
Dear Reviewer,
please find in the attached file the response to your comments.
Thank you for improving the quality of our work.
Best regards,
Fiammetta Berlinguer

Round 2
Reviewer 1 Report
PREAMBLE
As a first point and a general principle, the authors should learn to be polite to reviewers. Their response is bordering rudeness.
The manuscript does not present any novel concepts. The effects of energy on the reproductive cycle of ewes are well-established worldwide. The authors should read the papers of Russell in the 1970's, in order to fully appreciate how old this knowledge is. It is really distressing to see current scientists to ignore work of decades, in order to achieve a publication.
SPECIFIC COMMENTS
I wish to see the hard data regarding the findings presented in Figure 2. The authors can leave the figures in the main text, no problem there, but tables and hard data must be presented. Sofar, no hard data have been provided by the authors. As the findings are shown now, they are not convincing about the validity of the results. Further, the attempt to hide the hard data raises significant questions regarding their existence... For the sake of transparency, which is now pivotal throughout the scientific community, the authors MUST provide hard data and data in tables.
With regard to the ultrasonographic images, the following details are needed: type of probe, frequency, depth, technique, grey-scale intensity. Also, the authors must explain, why they presented images only with corporae lutae and not with follicles.
All in all, this is a manuscript that raises significant suspicions. The authors so far have not managed to answer the previous comments fully, but rather attacked the comments, rather that providing convincing answers. All the above do not provide definitive evidence regarding the work.
The manuscript is now borderline for rejection.
Reviewer 3 Report
In this revised version, the authors report some evidences that the ewes are in anestous and changed “…is as…” to “…can be…” in conclusion section (L506). I think that is enougth to mitigate the major issues described in the original version.
Nevertheless, in some minor points should be corrected before acceptance:
L52: The reference [2] was removed. Please update the citation´s number.
L57: This abreviation was defined in L166. Please correct. There are some troubles with the abreviations used in the document (e.g., NEFA was not defined; it is used as keyword). Please check all abreviations.
L124: I sugggest to chance the word “…hence…” to “…suggesting… “ or similar. Usually we confirm the seasonal anestrous by the absence of CL / low P4 levels during about 25-30 days consecutives. Ewes can be in transition period, short oestrus cycles (premature luteolysis) can occurs at this time, etc. so the term “…hence…” can be not enterily true.
L268-270: I think that you dont need to repeat all this information here. You can report that other than the ultrasound ovarian scannings in the pre-experimental period, …
Table 2, line 3: I suggest to begin with “a” in GLU group.